# Taller height and risk of coronary heart disease and cancer: A within-sibship Mendelian randomization study

Laurence J Howe[1,2,3]*, Ben Brumpton[3,4,5], Humaira Rasheed[1,3], Bjørn Olav Åsvold[3,6], George Davey Smith[1,2], Neil M Davies[1,2,3]*

[1]Medical Research Council Integrative Epidemiology Unit, Population Health Sciences, University of Bristol, Bristol, United Kingdom; [2]Population Health Sciences, Bristol Medical School, University of Bristol, Barley House, Oakfield Grove, Bristol, United Kingdom; [3]K.G. Jebsen Center for Genetic Epidemiology, Department of Public Health and Nursing, NTNU, Norwegian University of Science and Technology, Trondheim, Norway; [4]HUNT Research Center, Department of Public Health and Nursing, NTNU, Norwegian University of Science and Technology, Levanger, Norway; [5]Clinic of Medicine, St. Olavs Hospital, Trondheim University Hospital, Trondheim, Norway; [6]Department of Endocrinology, Clinic of Medicine, St. Olavs Hospital, Trondheim University Hospital, Trondheim, Norway

*For correspondence:
laurence.howe@bristol.ac.uk (LJH);
neil.davies@bristol.ac.uk (NMD)

Competing interest: The authors declare that no competing interests exist.

## Abstract

**Background:** Taller people have a lower risk of coronary heart disease but a higher risk of many cancers. Mendelian randomization (MR) studies in unrelated individuals (population MR) have suggested that these relationships are potentially causal. However, population MR studies are sensitive to demography (population stratification, assortative mating) and familial (indirect genetic) effects.

**Methods:** In this study, we performed within-sibship MR analyses using 78,988 siblings, a design robust against demography and indirect genetic effects of parents. For comparison, we also applied population MR and estimated associations with measured height.

**Results:** Within-sibship MR estimated that 1 SD taller height lowers the odds of coronary heart disease by 14% (95% CI: 3–23%) but increases the odds of cancer by 18% (95% CI: 3–34%), highly consistent with population MR and height-disease association estimates. There was some evidence that taller height reduces systolic blood pressure and low-density lipoprotein cholesterol, which may mediate some of the protective effects of taller height on coronary heart disease risk.

**Conclusions:** For the first time, we have demonstrated that the purported effects of height on adulthood disease risk are unlikely to be explained by demographic or familial factors, and so likely reflect an individual-level causal effect. Disentangling the mechanisms via which height affects disease risk may improve the understanding of the etiologies of atherosclerosis and carcinogenesis.

**Funding:** This project was conducted by researchers at the MRC Integrative Epidemiology Unit (MC_UU_00011/1) and also supported by a Norwegian Research Council Grant number 295989.

## Editor's evaluation

The authors examined the role of height in cancer, coronary heart disease and cardiovascular disease risk factors, using four different designs. They found that height increases risk of cancer and decreases risk of coronary heart disease, while the associations for the cardiovascular disease risk factors were largely null. This will be mainly of interest to epidemiologists.

## Introduction

Height is a classical complex trait influenced by genetic and early-life environmental factors. Despite the nonmodifiable nature of adult height, evaluating the effects of height on noncommunicable disease risk can give insights into the etiology of adulthood diseases (*Emerging Risk Factors Collaboration, 2012*; *Davey Smith et al., 2000*). Two major groupings of diseases, cardiovascular disease and cancer, have divergent associations with height (*Emerging Risk Factors Collaboration, 2012*; *Davey Smith et al., 2000*; *Stefan et al., 2016*). Taller people are less likely to develop cardiovascular disease, including coronary heart disease (CHD) (*Emerging Risk Factors Collaboration, 2012*; *Nelson et al., 2015*; *Nüesch et al., 2016*; *Hebert et al., 1993*; *Batty et al., 2009*; *Marouli et al., 2019*) and stroke (*Njølstad et al., 1996*), but more likely to be diagnosed with cancer (*Emerging Risk Factors Collaboration, 2012*; *Green et al., 2011*; *Zhang et al., 2015*; *Thrift et al., 2015*; *Dixon-Suen et al., 2018*; *Batty et al., 2006*; *Gunnell et al., 2001*; *Perkins et al., 2016*). The mechanisms via which height influences disease risks are unclear. The association between height and cardiovascular disease may be mediated via favorable lipid profiles (*Emerging Risk Factors Collaboration, 2012*; *Nelson et al., 2015*), lower systolic blood pressure (SBP) (*Emerging Risk Factors Collaboration, 2012*; *Langenberg et al., 2003*), lung function (*Marouli et al., 2019*; *Gunnell et al., 2003*), lower heart rate (*Smulyan et al., 1998*), and coronary artery vessel dimension (*O'Connor et al., 1996*). The increased cancer incidence among taller individuals could relate to early-life exposure to hormones such as insulin-like growth factor 1 (IGF-1) (*Renehan et al., 2004*; *Clayton et al., 2011*) or the increased number of cells in taller individuals (*Stefan et al., 2016*; *Green et al., 2011*; *Albanes and Winick, 1988*). However, although overall cancer risk is higher among taller individuals (*Green et al., 2011*; *Batty et al., 2006*; *Gunnell et al., 2001*), there is some evidence for heterogeneity across cancer subtypes with null or inverse associations observed between height and risk of stomach, oropharyngeal, and esophageal cancers (*Green et al., 2011*; *Batty et al., 2006*; *Gunnell et al., 2001*; *Perkins et al., 2016*).

Height is highly heritable, but the average height across the European populations has increased over the last hundred years (*Hatton, 2013*), illustrating the effects of early-life environmental factors such as nutrition and childhood infections. The associations of height with adulthood diseases and relevant biomarkers could reflect the biomechanical effects relating to increased stature (e.g., number of cells or larger arteries; *O'Connor et al., 1996*) or could reflect confounding by early-life environmental factors that influence both height and later-life health such as parental socioeconomic position. For example, wealthier parents may provide their offspring with better nutrition, leading to increased adult height, and a better education, potentially leading to improved health in adulthood (*Perkins et al., 2016*). Thus, it is unclear whether height has a causal effect on the risk of cardiovascular disease and cancer or if a confounding factor influences both height and disease risk.

Mendelian randomization (*Smith and Ebrahim, 2003*) analyses, using genetic variants associated with height as a proxy for observed height, have been used to strengthen the evidence for causal effects of height on adulthood diseases (*Nelson et al., 2015*; *Nüesch et al., 2016*; *Zhang et al., 2015*; *Thrift et al., 2015*; *Dixon-Suen et al., 2018*). The underlying premise being that genetic variants associated with height, unlike height itself, are unlikely to be associated with potential confounders such as childhood nutrition. However, there is growing evidence that estimates from genetic epidemiological studies using unrelated individuals may capture effects relating to demography (population stratification, assortative mating) and familial effects (e.g., indirect genetic effects of relatives where parental genotype influences offspring phenotypes) (*Barton and Hermisson, 2019*; *Berg et al., 2019*; *Sohail et al., 2019*; *Ruby et al., 2018*; *Haworth et al., 2019*; *Brumpton et al., 2019*; *Lee et al., 2018*; *Kong et al., 2018*; *Young et al., 2018*). Indeed, recent articles have illustrated the potential for genetic analyses of height to be affected by these biases (*Berg et al., 2019*; *Sohail et al., 2019*), including a Mendelian randomization study of height on education (*Brumpton et al., 2019*). One approach to overcome these potential biases is to use data from siblings (*Brumpton et al., 2019*; *Davies et al., 2019*) and exploit the shared early-life environment of siblings and the random segregation of alleles during meiosis (*Smith and Ebrahim, 2003*). Indeed, true Mendelian randomization was initially proposed as existing within a parent-offspring design (*Smith and Ebrahim, 2003*; *Davey Smith et al., 2020*; *Figure 1*).

Here, we used data from 40,275 siblings from UK Biobank (*Bycroft et al., 2018*) and 38,723 siblings from the Norwegian HUNT study (*Krokstad et al., 2013*) to estimate the effects of adulthood height on CHD, cancer risk, and relevant biomarkers. Study-level information is contained in *Table 1*. We

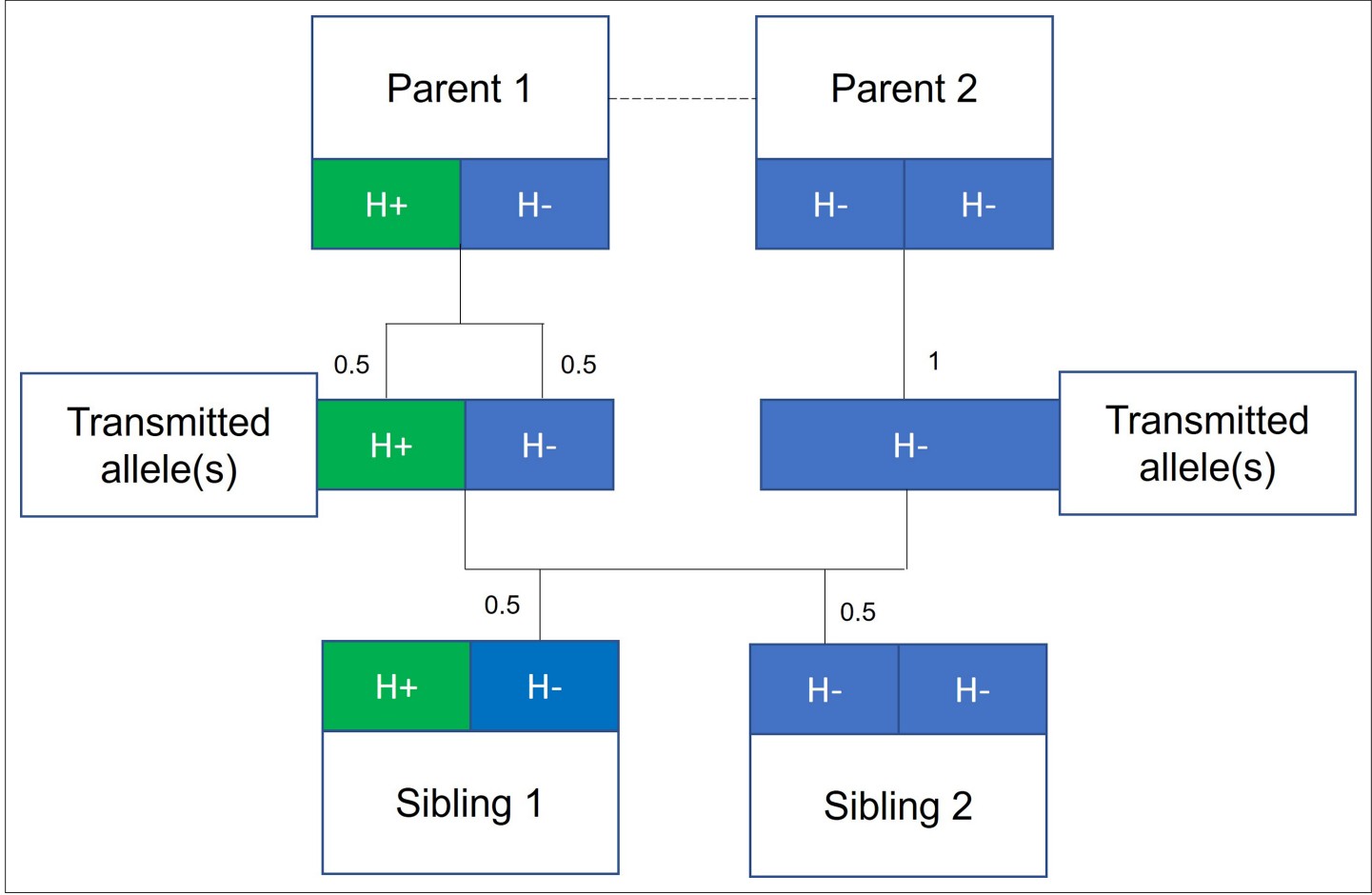

**Figure 1.** Mendelian randomization within families. The random allocation of alleles within a parent-offspring quad (two parents and two offspring), initially observed by Mendel, is illustrated. Consider a height influencing genetic variant H where on average individuals with the H+ allele are taller than individuals with the H- allele. From Mendel's law of segregation, parent 1, who is heterozygous at this allele, has an equal chance of transmitting either an H+ or H- allele to offspring. Parent 2, homozygous at this allele, will always transmit a copy of the H- allele. It follows that 50% of this pair's offspring will be heterozygous (as parent 1) and 50% will be homozygous for the H- allele (as parent 2). On average, the heterozygous offspring will be taller than the homozygous H- offspring, with this difference a consequence of random segregation of gametes.

report the estimates of the effects of height on CHD and cancer from both phenotypic models and Mendelian randomization, with and without accounting for family structure.

## Methods

### UK Biobank

#### Overview

UK Biobank is a large-scale prospective cohort study, described in detail previously (*Bycroft et al., 2018*; *Sudlow et al., 2015*). In brief, 503,325 individuals aged between 38 and 73 years were recruited between 2006 and 2010 from across the United Kingdom. For the purpose of this study, we used a subsample of 40,275 siblings from 19,588 families (*Brumpton et al., 2019*). Full-siblings were derived using UK Biobank-provided estimates of pairwise identical by state (IBS) kinships (>0.5–21 * IBS0, <0.7) and IBS0 (>0.001, <0.008), the proportion of unshared loci (*Hill and Weir, 2011*). This research has been conducted using the UK Biobank Resource under Application Number 15825. UK Biobank has ethical approval from the North West Multi-centre Research Ethics Committee (MREC). All UK Biobank participants provided written informed consent.

**Table 1.** UK Biobank and HUNT study characteristics.
Information on the UK Biobank and Norwegian HUNT studies, including descriptive of the sibling samples, is given.

| | UK Biobank | HUNT |
|---|---|---|
| Sibling sample:<br>N individuals (N sibships) | 40,275 (19,588) | 38,723 (15,179) |
| Recruitment period:<br>years | 2006–2010 | HUNT2 (1995–97)<br>HUNT3 (2006–08) |
| Year of birth:<br>median (Q1, Q3) | 1950 (1945, 1956) | 1951 (1937, 1963) |
| Sex:<br>male (%) | 42.2 | 48.7 |
| Male height (cm):<br>mean (SD) | 175.7 (6.7) | 177.6 (6.7) |
| Female height (cm):<br>mean (SD) | 162.4 (6.2) | 164.4 (6.3) |
| Coronary heart disease:<br>N cases (% of sample) | 3006 (7.5%) | 6447 (16.6%) |
| Cancer:<br>N cases (% of sample) | 6724 (16.5%) | 2323 (6.0%) |

## Phenotype data

At baseline, study participants attended an assessment center where they completed a touch-screen questionnaire, were interviewed, and had various measurements and samples taken. Height (field ID: 12144-0.0) and sitting height (field ID: 20015-0.0) were measured using a Seca 202 device at the assessment center. Seated height is equivalent to trunk length, leg length was defined as height minus seated height, and the leg to trunk ratio was calculated by taking the ratio of leg and trunk length. SBP was measured using an automated reading from an Omron Digital blood pressure monitor (field ID: 4080-0.0). Biomarkers of interest, including direct low-density lipoprotein cholesterol (LDL-C), high-density lipoprotein cholesterol (HDL-C), triglycerides (TG), glucose, and IGF-1, were measured using blood samples and the Beckman Coulter AU5800 or the DiaSorin LIASON XL (IGF-1) analyzers.

International Classification of Disease (10th edition) (ICD10) codes and Office of Population Censuses and Surveys Classifications of Interventions and Procedures (OPCS) codes were used to identify CHD and cancer (all subtypes and a stratified analysis) cases using several data sources: (1) secondary care data from Hospital Episode Statistics (HES), (2) death register data, and (3) cancer registry data. The stratified analysis included a subset of cancer subtypes (lung, oropharyngeal, stomach, esophageal, pancreatic, bladder, and multiple myeloma). Relevant codes are given in *Supplementary file 1A*. Both prevalent and incident cases were included in the analyses.

## Genotyping

The UK Biobank study participants (N = 488,377) were genotyped using the UK BiLEVE (N = 49,950) and the closely related UK Biobank Axiom Arrays (N = 438,427). Directly genotyped variants were pre-phased using SHAPEIT3 (*O'Connell et al., 2016*) and imputed using Impute4 and the UK10K (*Walter et al., 2015*), Haplotype Reference Consortium (*McCarthy et al., 2016*) and 1000 Genomes Phase 3 (*Genomes Project Consortium, 2015*) reference panels. More details are given in a previous publication (*Bycroft et al., 2018*).

## HUNT
### Overview

The Trøndelag Health Study (HUNT) is a series of general health surveys of the adult population of the demographically stable Nord-Trøndelag region, Norway, as detailed in a previous study (*Holmen et al., 2003*). The entire adult population of this region (~90,000 adults in 1995) is invited to attend a health survey (includes comprehensive questionnaires, an interview, clinical examination, and detailed

phenotypic measurements) every 10 years. To date, four health surveys have been conducted, HUNT1 (1984–1986), HUNT2 (1995–1997), HUNT3 (2006–2008), and HUNT4 (2017–2019), and all surveys have a high participation rate (*Krokstad et al., 2013*). This study includes 38,723 siblings from 15,179 families who participated in the HUNT2 and HUNT3 surveys. Siblings were identified using KING software (*Manichaikul et al., 2010*), with pairs defined as follows: kinship coefficient between 0.177 and 0.355, the proportion of the genomes that share two alleles identical by descent (IBD) > 0.08, and the proportion of the genome that share zero alleles IBD > 0.04. The use of HUNT data in this study was approved by the Regional Committee for Ethics in Medical Research, Central Norway (2017/2479). All HUNT study participants provided written informed consent.

## Phenotype data

Height was measured to the nearest 1.0 cm using standardized instruments with participants wearing light clothes without shoes. SBP was measured using automated oscillometry (Critikon Dinamap 845XT and XL9301, acquired by GE Medical Systems Information Technologies in 2000) on the right arm in a relaxed sitting position (*Holmen et al., 2003*; *Krokstad et al., 2013*). SBP was measured twice with a 1 min interval between measurement with the mean of both measurements used in this study.

All HUNT participants provided nonfasting blood samples when attending the screening site. Total cholesterol, HDL-C, and TG levels in HUNT2 were measured in serum samples using enzymatic colorimetric methods (Boehringer Mannheim, Mannheim, Germany). In HUNT3, participants' total cholesterol was measured by enzymatic cholesterol esterase methodology; HDL-C was measured by accelerator selective detergent methodology; and TGs were measured by glycerol phosphate oxidase methodology (Abbott, Clinical Chemistry, USA). LDL-C levels were calculated using the Friedewald formula (*Friedewald et al., 1972*) in both surveys. Participants in HUNT with TG levels ≥ 4.5 mmol/L (n = 1349) were excluded for LDL-C calculation as the Friedewald formula is not valid at higher TG levels. For all these phenotypes, if the participant attended both HUNT2 and HUNT3 surveys, then the values from HUNT2 were used for the analysis presented here.

The unique 11-digit identification number of every Norwegian citizen was used to link the HUNT participant records with the hospital registry, which included the three hospitals in the area (up to March 2019). We used ICD-10 and ICD-9 codes 410–414 and I20–I25 to define CHD, including both prevalent and incident cases. Cancer status (yes/no) was self-reported in HUNT2, HUNT3, and HUNT4 questionnaires. Individuals with discordant responses across different questionnaires were excluded from analyses. Due to the nature of cancer data collection, only prevalent cancer cases were included in analyses.

## Genotyping

DNA samples were available from 71,860 HUNT samples from HUNT2 and HUNT3 and were genotyped (*Krokstad et al., 2013*) using one of the three different Illumina HumanCoreExome arrays: HumanCoreExome12 v1.0 (n = 7570), HumanCoreExome12 v1.1 (n = 4960), and University of Michigan HUNT Biobank v1.0 (n = 58,041; *HumanCoreExome-24* v1.0, with custom content). Quality control was performed separately for genotype data from different arrays. The call rate of genotyped samples was >99%. Imputation was performed on samples of recent European ancestry using Minimac3 (v2.0.1, http://genome.sph.umich.edu/wiki/Minimac3) (*Das et al., 2016*) from a merged reference panel constructed from (1) the Haplotype Reference Consortium panel (release version 1.1) (*McCarthy et al., 2016*) and (2) a local reference panel based on 2202 whole-genome sequenced HUNT participants (*Zhou et al., 2017*). The subjects included in the study were of European ancestry and had passed the quality control.

## Statistical analysis

### Population and within-sibship models

The population model is a conventional regression model where the outcome is regressed (linear or logistic) against the exposure (height or height polygenic score [PGS]) with the option to include covariates.

The within-sibship model is an extension to the population model that includes a family mean term, the average exposure value across each family (height or height PGS), with each individual exposure value centered about the family mean exposure. To account for relatedness between siblings,

standard errors are clustered by family in both models. More information on these models is contained in previous publications (*Brumpton et al., 2019*; *Howe et al., 2021*) with statistical code available on GitHub (*Howe, 2022*).

## Phenotypic and Mendelian randomization analyses

In phenotypic analyses, we used regression models (within-sibship and population) to estimate the association between measured height and all outcomes (CHD, cancer, SBP, LDL-C, HDL-C, TG, glucose, and IGF-1) using linear models for continuous outcomes and logistic models for binary disease outcomes. In both cohorts, we used a standardized measure of height after adjusting for age and sex and also standardized continuous outcomes after adjusting for age and sex.

In Mendelian randomization analyses, we fit regression models as above but used an age/sex-standardized height PGS instead of measured height. The height PGS was constructed in PLINK (*Purcell et al., 2007*) using 372 independent (LD clumping: 250 kb, r2 < 0.01, p<5 × 10$^{-8}$) genetic variants from a previous height genome-wide association study (*Wood et al., 2014*) that did not include UK Biobank or HUNT. Again, we standardized and adjusted for age/sex for continuous outcomes. To estimate the effect of the PGS on height, we fit a model regressing measured standardized height against the height PGS. We then generated scaled Mendelian randomization estimates by taking the Wald ratio of the PGS-outcome associations and the PGS-height associations. All statistical analyses were conducted using R (v. 3.5.1).

There are three core instrumental variable assumptions for Mendelian randomization analyses. First, the genetic variants should be robustly associated with the exposure (relevance). Second, there should be no unmeasured confounders of the genetic variant-outcome association (independence). Third, the genetic variants should only influence the outcome via their effect on the exposure (the exclusion restriction) (*Haycock et al., 2016*; *Didelez and Sheehan, 2007*; *Lawlor et al., 2008*).

## UK Biobank and HUNT meta-analyses

We performed phenotypic and Mendelian randomization analyses (using population and within-sibship models) in both UK Biobank and HUNT. For phenotypes measured in both studies (CHD, cancer, LDL-C, HDL-C, TG), we combined estimates across both studies using a fixed-effects model in the metafor R package for meta-analysis. We tested for heterogeneity between UK Biobank/HUNT estimates using the difference of two means test statistic (*Altman and Bland, 2003*).

## Outcomes

Using the previously described models and meta-analysis procedure, we estimated the effects of height on CHD, cancer, LDL-C, HDL-C, TG, glucose, and IGF-1. As a sensitivity analysis, we used phenotypic models to evaluate the associations between dimensions of height (leg length, trunk length, and leg to trunk ratio) with CHD and cancer in UK Biobank. A further sensitivity analysis involved repeating cancer analyses in UK Biobank with a subset of cancers not phenotypically associated with height (described above).

# Results

## Adulthood height and risk of CHD and cancer

We found consistent evidence across population and within-sibship models, using both measured height and a height PGS, that taller adulthood height reduced CHD risk and increased the risk of cancer (*Supplementary file 1B and C*).

Within-sibship Mendelian randomization estimated that 1 SD taller height (approximately 6.8 cm for men and 6.2 cm for women) reduced the odds of CHD by 14% (95% CI 3–23%) but increased the odds of cancer by 18% (95% CI 3–34%). These estimates were consistent across analyses using measured height as well as with population Mendelian randomization estimates. For example, population Mendelian randomization analyses estimated that 1 SD taller height reduced the odds of CHD by 10% (95% CI 4–16%) and increased the odds of cancer by 9% (95% CI 2–16%) (*Table 2*, *Figure 2*).

We then evaluated the associations between dimensions of height (trunk length, leg length, and leg to trunk ratio) and risk of CHD/cancer in UK Biobank. We found little evidence of heterogeneity between estimates, although stronger conclusions are limited by statistical power (*Supplementary*

**Table 2.** Mendelian randomization (MR) results: change in outcome (SD units), per 1 SD increase in height. Population and within-sibship MR estimates of height on the eight different outcomes are shown. The presented estimates are from UK Biobank, HUNT, and the combined fixed-effects meta-analysis. The heterogeneity p-value refers to the difference between the UK Biobank and HUNT estimates.

| Outcome | Model | UK Biobank | HUNT | Combined | Study heterogeneity p-value |
|---|---|---|---|---|---|
| Systolic blood pressure | Population | −0.044 (−0.074, −0.015) | −0.025 (−0.057, 0.008) | −0.036 (−0.058, −0.014) | 0.38 |
| | Within-sibship | −0.077 (−0.137, −0.017) | 0.010 (−0.040, 0.059) | −0.025 (−0.063, 0.013) | 0.03 |
| High-density lipoprotein cholesterol | Population | −0.039 (−0.070, −0.008) | −0.010 (−0.043, 0.024) | −0.025 (−0.048, −0.003) | 0.21 |
| | Within-sibship | −0.038 (−0.096, 0.021) | 0.001 (−0.046, 0.047) | −0.014 (−0.050, 0.022) | 0.31 |
| Low-density lipoprotein cholesterol | Population | −0.066 (−0.095, −0.036) | −0.065 (−0.098, −0.032) | −0.065 (−0.087, −0.044) | 0.99 |
| | Within-sibship | −0.083 (−0.141, −0.025) | −0.014 (−0.061, 0.033) | −0.041 (−0.078, −0.005) | 0.07 |
| Triglycerides | Population | 0.011 (−0.018, 0.040) | −0.006 (−0.038, 0.027) | 0.004 (−0.018, 0.025) | 0.43 |
| | Within-sibship | 0.024 (−0.034, 0.081) | 0.032 (−0.018, 0.082) | 0.028 (−0.009, 0.066) | 0.84 |
| Glucose | Population | 0.032 (0.005, 0.060) | N/A | 0.032 (0.005, 0.060) | N/A |
| | Within-sibship | 0.023 (−0.030, 0.077) | N/A | 0.023 (−0.030, 0.077) | N/A |
| IGF-1 | Population | −0.005 (−0.035, 0.025) | N/A | −0.005 (−0.035, 0.025) | N/A |
| | Within-sibship | −0.045 (−0.093, 0.004) | N/A | −0.045 (−0.093, 0.004) | N/A |
| Cancer (OR) | Population | 1.12 (1.04, 1.20) | 0.99 (0.88, 1.12) | 1.09 (1.02, 1.16) | 0.089 |
| | Within-sibship | 1.21 (1.03, 1.42) | 1.12 (0.90, 1.39) | 1.18 (1.03, 1.34) | 0.57 |
| Coronary heart disease (OR) | Population | 0.94 (0.84, 1.04) | 0.88 (0.80, 0.96) | 0.90 (0.84, 0.96) | 0.33 |
| | Within-sibship | 0.81 (0.65, 1.02) | 0.88 (0.77, 1.00) | 0.86 (0.77, 0.97) | 0.55 |

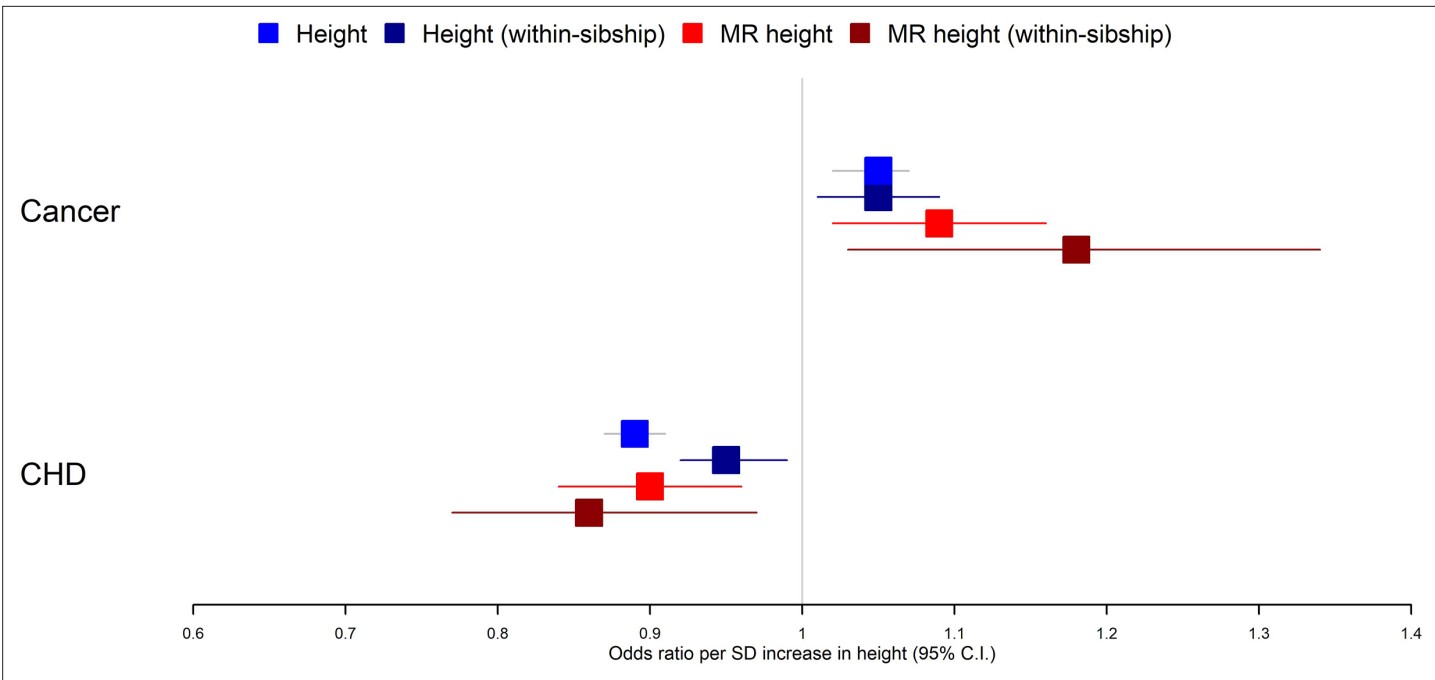

**Figure 2.** Taller height and risk of coronary heart disease and cancer. The meta-analysis results from four different models used to evaluate the effect of height on coronary heart disease (CHD) and cancer risk are displayed. First, a phenotypic population model with measured height as the exposure and age and sex included as covariates. Second, a within-sibship phenotypic model with the family mean height included as an additional covariate to account for family structure. Third, a population Mendelian randomization model with height polygenic score (PGS) as the exposure exploiting advantageous properties of genetic instruments. Fourth, a within-sibship Mendelian randomization model with the family mean PGS included as a covariate to control for parental genotypes. Across all four models, we found consistent evidence that taller height reduces the odds of CHD and increases the odds of cancer.

*file 1D*). We also ran a sensitivity analysis in UK Biobank, rerunning height-cancer analyses including only cases with one of seven cancer subtypes (lung, oropharyngeal, stomach, esophageal, pancreatic, bladder, and multiple myeloma) for which a previous study found little evidence they associated with height (*Green et al., 2011*). These subtypes generally show very strong social patterning, which could explain the attenuated associations with height that is also often socially patterned. As expected, the association of measured height with this subset of cancers (population OR 0.99; 95% CI 0.92–1.06; within-sibship OR 1.01; 95% CI 0.88–1.15) was less strong than the association between height and the all-cancer outcome (population OR 1.05; 95% CI 1.02–1.07; within-sibship OR 1.05; 95% CI 1.01–1.09). Mendelian randomization estimates were imprecise because of the modest number of cases for these cancers (*Supplementary file 1E*).

## Adulthood height and biomarkers

Using measured biomarkers, both population and within-sibship models found evidence for the association between taller height and lower SBP, lower circulating LDL-C, and higher circulating IGF-1 levels. There was some evidence for heterogeneity in phenotypic associations between height and biomarkers in UK Biobank and HUNT, such as for SBP, which was more strongly associated with height in UK Biobank (*Supplementary file 1B*).

Population Mendelian randomization results suggested that taller height reduced SBP (per 1 SD taller height, 0.036 SD decrease; 95% CI 0.014–0.058), LDL-C (per 1 SD taller height, 0.065 SD decrease; 95% CI 0.044–0.087), HDL-C (per 1 SD taller height, 0.025 SD decrease; 95% CI 0.003–0.048) but increased glucose (per 1 SD taller height, 0.032 SD increase; 95% CI 0.005–0.060). In contrast, we found little evidence that taller height affected TG or IGF-1 levels. Within-sibship Mendelian randomization estimates were consistent with population estimates; SBP (per 1 SD taller height, 0.025 SD decrease; 95% CI –0.013 to 0.063), LDL-C (per 1 SD taller height, 0.041 SD decrease; 95% CI 0.005–0.078), HDL-C (per 1 SD taller height, 0.014 SD decrease; 95% CI –0.022 to 0.050) and glucose (per 1 SD taller height, 0.023 SD increase; 95% CI –0.030 to 0.077) (*Figure 3*, *Table 2*).

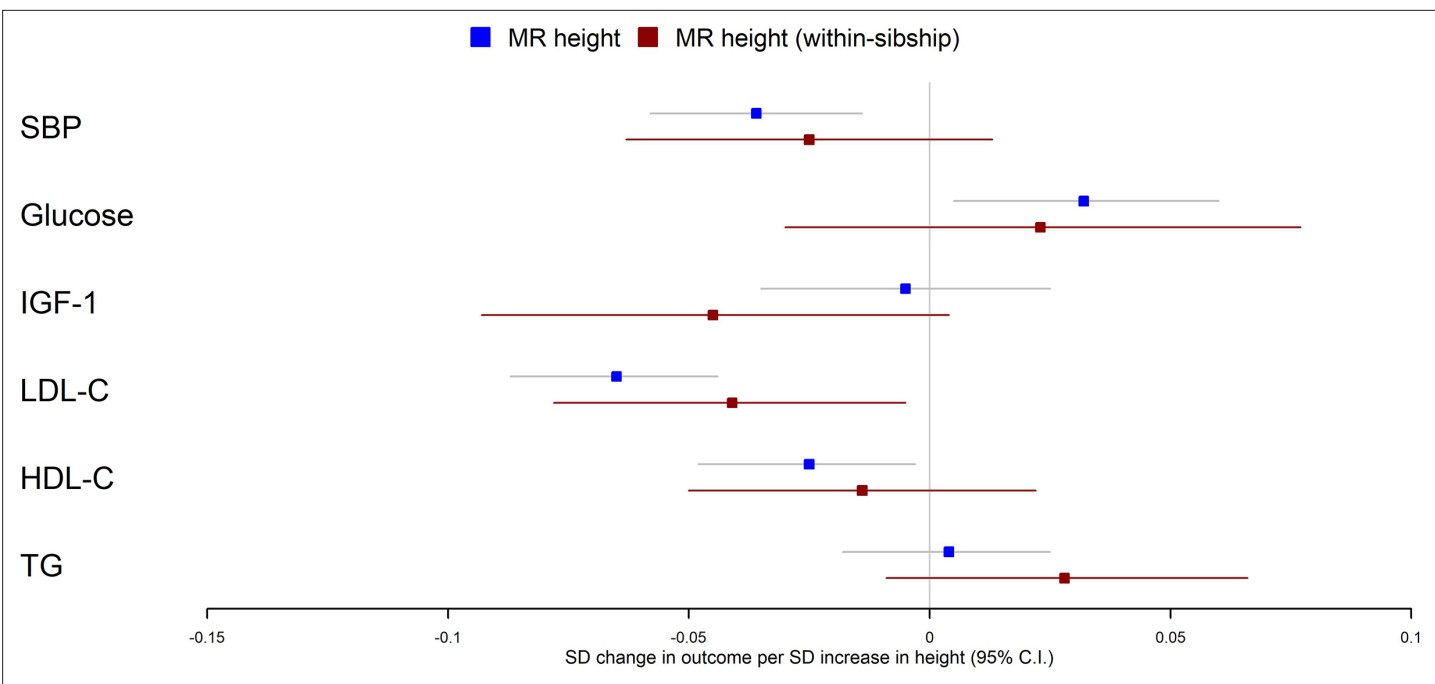

**Figure 3.** Mendelian randomization estimates of the effects of taller height on biomarkers. The meta-analysis results from population and within-sibship Mendelian randomization analyses estimating the effect of taller height on biomarkers across UK Biobank and HUNT are shown. The estimates were broadly similar between the two models, suggesting the modest effects of demography and indirect genetic effects.

There was some putative evidence for heterogeneity in the Mendelian randomization effect estimates between UK Biobank and HUNT. For example, within-sibship Mendelian randomization estimate suggested the effects of height on SBP in UK Biobank (0.077 SD decrease; 95% CI 0.017–0.137) but the effect estimate was in the opposite direction in HUNT (0.010 SD increase; 95% CI –0.040 to 0.059; heterogeneity p=0.03) (*Table 2*).

## Discussion

In this study, we used sibling data from two large biobanks to estimate the effects of height on CHD, cancer, and relevant biomarkers. We found consistent evidence across all models, including within-sibship Mendelian randomization, that taller height is protective against CHD but increases the risk of cancers. We found less consistent evidence for the effects of height on biomarkers; population and within-sibship phenotypic models as well as population Mendelian randomization models suggested modest effects of taller height on SBP, LDL-C, and HDL-C. However, the confidence intervals for within-family Mendelian randomization of height and biomarkers were too wide to draw strong conclusions.

Our findings are largely consistent with previous studies (*Emerging Risk Factors Collaboration, 2012*; *Nelson et al., 2015*; *Nüesch et al., 2016*; *Hebert et al., 1993*; *Marouli et al., 2019*; *Green et al., 2011*; *Zhang et al., 2015*; *Thrift et al., 2015*; *Dixon-Suen et al., 2018*; *Carslake et al., 2013*) that used nonsibling designs, and with the hypothesis that height affects CHD and cancer risk. However, previous studies were potentially susceptible to bias relating to geographic and socioeconomic variation in height and height genetic variants (*Barton and Hermisson, 2019*; *Sohail et al., 2019*; *Lee et al., 2018*). Indeed, a recent within-sibship Mendelian randomization study found that the previously reported effects of height and body mass index on educational attainment were greatly attenuated when using siblings (*Brumpton et al., 2019*). Here, we provided robust evidence for individual-level effects of height by demonstrating that the previous evidence for effects of height on adulthood disease risk is unlikely to have been confounded by demography or indirect genetic effects. The major strengths of our work are the use of within-sibship Mendelian randomization (*Davies et al., 2019*) and the triangulation (*Lawlor et al., 2016*) of evidence from across phenotypic, genetic, and within-sibship models.

A limitation of our analyses is that because of limited sibling data and the statistical inefficiency of within-family models, we have limited statistical power to investigate the effects of height on disease subtypes,further explore the mechanisms using multivariable Mendelian randomization (***Burgess and Thompson, 2015***), and perform sensitivity analyses to evaluate horizontal pleiotropy. An additional limitation is that our study may have been susceptible to selection and survival biases relating to nonrandom participation in UK Biobank and/or HUNT and the requirement of at least two siblings to survive to be recruited. Indeed, cancer and CHD are both leading worldwide causes of mortality and so cases for one disease may have a reduced likelihood of developing the other disease due to increased mortality. Therefore, our study may have been susceptible to survival bias relating to competing risks. We mitigated this by defining cases for both diseases using both nonfatal and fatal events. Our study analyzed families with two or more siblings jointly participating in a cohort; nevertheless, further research is required to investigate the impact of selection bias on family studies.

Adulthood height is nonmodifiable, and the interpretation of causality is nuanced because it is unclear whether biological effects relate to stature itself, increased childhood growth, or to factors highly correlated with height such as lung function (***Marouli et al., 2019***; ***Gunnell et al., 2003***) and artery length (***Palmer et al., 1990***). Previous studies ***Gunnell et al., 2001***; ***Langenberg et al., 2003***; ***Gunnell et al., 2003***; ***Regnault et al., 2014*** have explored the possibility that associations may relate to dimensions of height, with evidence that blood pressure is associated with trunk but not leg length (***Regnault et al., 2014***). Here, we found that the effects of height on disease risk due to leg or trunk length were similar. We found consistent effects of increased height across etiologically heterogeneous cancer subtypes, which implies that the mechanism could relate to the larger number of cells in taller individuals or a generalized growth phenotype. Notably there is minimal evidence of a correlation between the size of an organism and cancer risk (Peto's paradox), suggesting that the number of cells hypothesis could influence cancer risk in humans but would not explain variation in cancer risk across different organisms (***Caulin and Maley, 2011***). Our Mendelian randomization estimates for the effects of height on a subset of cancers not strongly phenotypically associated with height (***Green et al., 2011***) were consistent with the combined cancer estimates, although we had limited power in this dataset because of the modest prevalence of the cancer subtypes.

The estimated effects of height on disease risk were relatively consistent between the Norwegian HUNT and UK Biobank studies. Contrastingly, the heterogeneity between UK Biobank and HUNT for analyses involving SBP and LDL-C suggests that some effects of height could be population specific. Alternatively, heterogeneity could relate to the variance in associations between adulthood height and early-life environmental confounders across countries (***Perkins et al., 2016***). Additional explanations could relate to the differences in biomarker measurement between studies (e.g., measuring LDL-C directly or using the Friedewald formula, differences in fasting level before samples were taken), selection bias (***Munafò et al., 2016***), or differences between the cohorts in terms of recruitment and participation. Further work is required to investigate if our findings generalize to non-European populations; biological mechanisms could be expected to be largely consistent across populations but context-specific (e.g., social) mechanisms could lead to geographic heterogeneity.

To conclude, using within-sibship Mendelian randomization, we showed that height has individual-level effects on risk of CHD and cancers as well as several biomarkers. Larger family datasets and additional analyses including two-step (***Relton and Davey Smith, 2012***) and multivariable Mendelian randomization (***Burgess and Thompson, 2015***) could be used to investigate the potential mediators of these relationships.

## Acknowledgements

Quality Control filtering of the UK Biobank data was conducted by R Mitchell, G Hemani, T Dudding, and L Paternoster as described in the published protocol (https://doi.org/10.5523/bris.3074krb6t2fr j29yh2b03x3wxj). The University of Bristol support the MRC Integrative Epidemiology Unit (MC_UU_00011/1). NMD was supported by a Norwegian Research Council grant number 295989. The Trøndelag Health Study (The HUNT Study) is a collaboration between HUNT Research Centre (Faculty of Medicine and Health Sciences, NTNU, Norwegian University of Science and Technology), Trøndelag County Council, Central Norway Regional Health Authority, and the Norwegian Institute of Public Health. The funders had no role in study design, data collection and analysis, decision to publish, or

preparation of the manuscript. This publication is the work of the authors, who serve as the guarantors for the contents of this paper.

## Additional information

### Funding

| Funder | Grant reference number | Author |
|---|---|---|
| Norwegian Research Council | 295989 | Neil Martin Davies |
| MRC Integrative Epidemiology Unit | MC_UU_00011/1 | Laurence J Howe<br>Neil M Davies<br>George Davey Smith |

The funders had no role in study design, data collection and interpretation, or the decision to submit the work for publication.

### Author contributions

Laurence J Howe, Conceptualization, Formal analysis, Investigation, Methodology, Software, Writing – original draft, Writing – review and editing; Ben Brumpton, Data curation, Funding acquisition, Supervision, Writing – review and editing; Humaira Rasheed, Data curation, Writing – review and editing; Bjørn Olav Åsvold, Data curation, Funding acquisition, Writing – review and editing; George Davey Smith, Funding acquisition, Methodology, Supervision, Writing – review and editing; Neil M Davies, Funding acquisition, Investigation, Methodology, Supervision, Writing – review and editing

### Author ORCIDs

Laurence J Howe ⓘ http://orcid.org/0000-0002-2819-9686
Humaira Rasheed ⓘ http://orcid.org/0000-0002-3331-5864
George Davey Smith ⓘ http://orcid.org/0000-0002-1407-8314
Neil M Davies ⓘ http://orcid.org/0000-0002-2460-0508

### Ethics

Human subjects: This research has been conducted using the UK Biobank Resource under Application Number 15825. UK Biobank has ethical approval from the North West Multi-centre Research Ethics Committee (MREC). All UK Biobank participants provided written informed consent. The use of HUNT data in this study was approved by the Regional Committee for Ethics in Medical Research, Central Norway (2017/2479). All HUNT study participants provided written informed consent.

### Decision letter and Author response

Decision letter https://doi.org/10.7554/eLife.72984.sa1
Author response https://doi.org/10.7554/eLife.72984.sa2

## Additional files

### Supplementary files

• Supplementary file 1. Supplementary tables. (a) contains the different variables and codes used to define coronary heart disease, all cancers, and subset of cancer disease outcomes in UK Biobank. (b) contains population and within-sibship associations between measured height and the eight different outcomes. Presented estimates are from UK Biobank, HUNT, and the combined fixed-effects meta-analysis. The heterogeneity p-value refers to the difference between the UK Biobank and HUNT estimates. (c) contains population and within-sibship associations between the height polygenic score (PGS) and the eight different outcomes. Estimates are presented from UK Biobank and HUNT. (d) contains population and within-sibship associations between three height-related measures (leg length, trunk length, and leg/trunk ratio) and cancer and coronary heart disease from UK Biobank. Height associations are presented for comparison. (e) contains population and within-sibship phenotypic and Mendelian randomization estimates for height on a set of cancer subtypes with limited prior evidence for association with height. Estimates for the 'all cancers' outcome are

included for comparison. This analysis was conducted in UK Biobank only.

• Transparent reporting form

## Data availability

We used individual level data from the UK Biobank and HUNT cohorts. Participants in these studies have consented to the use of their data in medical research and so these data are not publicly available. Data access can be applied for by qualified researchers. For access to UK Biobank individual level participant data, please send enquiries to access@ukbiobank.ac.uk and see information on the UK Biobank website http://www.ukbiobank.ac.uk. UK Biobank access generally involves submitting project proposals which are evaluated by the study data access committee. Researchers associated with Norwegian research institutes can apply for the use of HUNT data and samples with approval by the Regional Committee for Medical and Health Research Ethics. HUNT data is governed by Norwegian law, therefore researchers from other countries may apply if collaborating with a Norwegian Principal Investigator. Detailed information on the data access procedure of HUNT can be found at https://www.ntnu.edu/hunt/data. Statistical code for population and within-sibship models used in the manuscript is available on GitHub https://github.com/Laurence-Howe/WithinSibshipModels/ (copy archived at swh:1:rev:44d2435d841bc424b56eee2d8534d52d d4adf763).

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
