## [Editor Report]

The authors examined the role of height in cancer, coronary heart disease and cardiovascular disease risk factors, using four different designs. They found that height increases risk of cancer and decreases risk of coronary heart disease, while the associations for the cardiovascular disease risk factors were largely null. This will be mainly of interest to epidemiologists.

---

## [Decision Letter]

**Decision letter after peer review:**

Thank you for submitting your article "Taller height and risk of coronary heart disease and cancer: a within-sibship Mendelian randomization study" for consideration by *eLife*. Your article has been reviewed by 2 peer reviewers, and the evaluation has been overseen by a Reviewing Editor and a Senior Editor. The following individual involved in review of your submission has agreed to reveal their identity: Sara Hägg (Reviewer #2).

The reviewers have discussed their reviews with one another, and the Reviewing Editor has drafted this letter to help you prepare a revised submission.

Essential revisions:

1. A major source of bias in studies of cause and effect is selection (or collider) bias. The study also needs to consider possible selection bias from inevitably only selecting survivors to recruitment of their height (measured or genetically endowed), the disease of interest and any competing risk of the disease of interest. Whether, the sibling design, by requiring at least two siblings to survive to recruitment, is more open to selection bias than a population-based design could also be considered.

2. The study provides an interesting comparison of four different methods. Height increasing cancer risk is plausible because of the potential mechanism given, i.e., IGF1, the consistency with well accepted biological theories from evolutionary biology, and that cancer deaths tend to occur at younger ages than cardiovascular disease deaths. The results for coronary artery disease are possible but, could be overstated Greater consideration should be also be given to as to how "an individual-level causal effect" observed largely in Europeans generalizes.

3. Whether the findings for height decreasing the risk of coronary heart disease could be partly due to an artefact of survivorship bias (https://en.wikipedia.org/wiki/Survivorship_bias) arising from taller people dying of cancer before they have a chance to get coronary heart disease should be considered in more detail.

4. Please explain more clearly about adjustment for age, does it account for the fact that risk of disease goes up exponentially with age? Please also consider whether cohort is also relevant. Cohort may affect both height and disease risk. In the UK Biobank the recruitment period is quite short, so cohort should not be an issue. Please clarify over what time period the HUNT study was recruited and whether cohort should be considered as well as age.

5. Adhere to *eLife* title standards. Also, not clear that both observational and MR estimates are calculated on individual level data from HUNT and UKB. I think the title should be modified and make more justice to the study.

Same thing with abstract. Lots of background added but not the phenotypic associations. By the way, why call it phenotypic- the usual phrase is observational association? Biomarker results not described either.

6. Add more discussion on the CVD-cancer paradox that has been discussed in literature.

*Reviewer #1 (Recommendations for the authors):*

Methods

Please explain more clearly about adjustment for age, does it account for the fact that risk of disease goes up exponentially with age? Please also consider whether cohort is also relevant. Cohort may affect both height and disease risk. In the UK Biobank the recruitment period is quite short, so cohort should not be an issue. Please clarify over what time period the HUNT study was recruited and whether cohort should be considered as well as age.

Is a p-value cutoff given for SNP selection into the height PGS?

Please clarify whether prevalent cases were included as well as incident cases. Given, genetics are a lifelong exposure, it might be better to include prevalent cases, so as to avoid time related biases, as explained here[1].

Results

Please give a table describing the baseline characteristics of the two studies, particularly in terms of age and sex, and possibly recruitment period.

Discussion

Please consider the limitations of this study in more detail, specifically.

1. Using a PGS for the exposure increases power but means it is difficult to check for any pleotropic effects. How might this affect interpretation of the study?

2. This study is carefully designed to avoid use of overlapping samples for exposure and outcome. Is the study used for height unbiased?

3. The possibility of selection bias from survivorship. This study only includes people who have survived to recruitment, so it will be missing people who died before recruitment because of their height or height-related genetic endowment, the disease of interest or a competing risk of the disease of interest. Missing these people can attenuate or reverse estimates for exposures that affect survival. If cancer causes death at earlier ages than coronary heart disease then the results for coronary heart disease could be an artefact of the competing risk of previous death from cancer.

4. Please clarify whether the results presented are specific to the UK Biobank and HUNT studies or explain their generalizability and transportability.

5. Could differences between UK Biobank and HUNT be due to differences to design?

6. Are both the comparisons between the sibling and population estimates for cancer and coronary heart disease in the expected directions?

[1] Hernán MA, Sauer BC, Hernández-Díaz S, Platt R, Shrier I: Specifying a target trial prevents immortal time bias and other self-inflicted injuries in observational analyses. J Clin Epidemiol 2016, 79:70-75.

*Reviewer #2 (Recommendations for the authors):*

How likely is it that the earlier MR results are misleading? Biased by demographics etc? It relates to the novelty of the paper because the MR association has been shown before multiple times.

What is the rationale for doing the biomarker analysis? Would it not be more informative to investigate if they mediate the effect from height on CHD?

Table 1 should be sample characteristics in HUNT and UKB.

What about sex stratifications? Is it possible to do for same-sex sib pairs? Is it of importance in the population at large regardless of sib ship pairs? For the traits used here it should matter.

All figures and tables should contain information so that they are stand alone items.

---

## [Author Response]

Essential revisions:1. A major source of bias in studies of cause and effect is selection (or collider) bias. The study also needs to consider possible selection bias from inevitably only selecting survivors to recruitment of their height (measured or genetically endowed), the disease of interest and any competing risk of the disease of interest. Whether, the sibling design, by requiring at least two siblings to survive to recruitment, is more open to selection bias than a population-based design could also be considered.

We have added the possibility of selection (collider) bias to the limitations section of the discussion. We have also noted that further work is required to understand the susceptibility of family studies to selection bias.

“An additional limitation is that our study may have been susceptible to selection and survival biases relating to non-random participation in UK Biobank and/or HUNT and the requirement of at least two siblings to survive to be recruited. Indeed, cancer and coronary heart disease are both leading worldwide causes of mortality and so cases for one disease may have reduced likelihood of developing the other disease due to increased mortality. Therefore, our study may have been susceptible to survival bias relating to competing risks. We mitigated this by defining cases for both diseases using both non-fatal and fatal events. Our study analysed families with two or more siblings jointly participating in a cohort; nevertheless, further research is required to investigate the impact of selection bias on family studies.”

2. The study provides an interesting comparison of four different methods. Height increasing cancer risk is plausible because of the potential mechanism given, i.e., IGF1, the consistency with well accepted biological theories from evolutionary biology, and that cancer deaths tend to occur at younger ages than cardiovascular disease deaths. The results for coronary artery disease are possible but, could be overstated Greater consideration should be also be given to as to how "an individual-level causal effect" observed largely in Europeans generalizes.

We have added to the discussion how our conclusions are based on individuals of recent European descent and so may not generalise to non-European populations if the mechanisms are context-specific.

“Further work is required to investigate if our findings generalise to non-European populations; biological mechanisms could be expected to be largely consistent across populations but context-specific (e.g., social) mechanisms could lead to geographical heterogeneity.”

3. Whether the findings for height decreasing the risk of coronary heart disease could be partly due to an artefact of survivorship bias (https://en.wikipedia.org/wiki/Survivorship_bias) arising from taller people dying of cancer before they have a chance to get coronary heart disease should be considered in more detail.

We have added discussion of competing risks between the two diseases to the discussion.

“An additional limitation is that our study may have been susceptible to selection and survival biases relating to non-random participation in UK Biobank and/or HUNT and the requirement of at least two siblings to survive to be recruited. Indeed, cancer and coronary heart disease are both leading worldwide causes of mortality and so cases for one disease may have reduced likelihood of developing the other disease due to increased mortality. Therefore, our study may have been susceptible to survival bias relating to competing risks. We mitigated this by defining cases for both diseases using both non-fatal and fatal events. Our study analysed families with two or more siblings jointly participating in a cohort; nevertheless, further research is required to investigate the impact of selection bias on family studies.”

4. Please explain more clearly about adjustment for age, does it account for the fact that risk of disease goes up exponentially with age? Please also consider whether cohort is also relevant. Cohort may affect both height and disease risk. In the UK Biobank the recruitment period is quite short, so cohort should not be an issue. Please clarify over what time period the HUNT study was recruited and whether cohort should be considered as well as age.

We adjusted for age (birth year / birth cohort) to capture phenotypic variation in both height and the disease/biomarker outcomes (disease risk increasing with age). Age / birth year is a potential confounder of the relationship between measured height and an outcome. However, age/birth year is very unlikely to confound the relationship between height genetic variants and an outcome because these factors cannot plausibly influence genotype. Selection on height could influence allele frequencies over time leading to variation by birth year but any changes in the context of this study would be minimal.

Similarly, cohort (HUNT 2 or HUNT 3) could be a potential confounder of the phenotypic analyses but is very unlikely to confound the MR analyses. The primary focus of our manuscript is on the within-sibship MR analyses and the triangulation of evidence across different study designs. We generally found consistent effect estimates across both phenotypic and MR analyses suggesting that non-linear age and cohort effects are unlikely to have explained our results.

At the request of the reviewers, we have added Table 1 which contains information on time periods of recruitment and other study-level information.

5. Adhere to eLife title standards. Also, not clear that both observational and MR estimates are calculated on individual level data from HUNT and UKB. I think the title should be modified and make more justice to the study.

The *eLife* guide to authors requires titles to be less than 120 characters, gives the following example:

Predicting the effect of statins on cancer risk using genetic variants from a Mendelian randomization study in the UK Biobank (125 characters)

We have removed the colon from our title to read:

“Taller height and risk of coronary heart disease and cancer, a within-sibship Mendelian randomization study” (107 characters)

We cannot include the study names (UK Biobank and HUNT) because the title would become too long. However, we are happy to consider alternative titles.

Same thing with abstract. Lots of background added but not the phenotypic associations. By the way, why call it phenotypic- the usual phrase is observational association? Biomarker results not described either.

We previously did not mention the phenotypic associations in the abstract in the interests of word count and brevity. We have now shortened the background section and added information on the phenotypic associations.

We used “phenotypic” over “observational” to describe the non-genetic analyses because the term “observational” is somewhat misleading because genetic data is also observational. We believe “phenotypic” makes it clearer that the analysis is non-genetic and only using phenotype data. However, we are happy to modify the terminology to “observational” at the discretion of the editors.

6. Add more discussion on the CVD-cancer paradox that has been discussed in literature.

Peto’s paradox notes that cancer risk correlates with body size within a species but cancer risk between species does not correlate with body size. We have added discussion of this to the manuscript.

“Notably there is minimal evidence of a correlation between the size of an organism and cancer risk (Peto’s paradox) suggesting that the number of cells hypothesis could influence cancer risk in humans but would not explain variation in cancer risk across different organisms ^63^.”

Reviewer #1 (Recommendations for the authors):MethodsPlease explain more clearly about adjustment for age, does it account for the fact that risk of disease goes up exponentially with age? Please also consider whether cohort is also relevant. Cohort may affect both height and disease risk. In the UK Biobank the recruitment period is quite short, so cohort should not be an issue. Please clarify over what time period the HUNT study was recruited and whether cohort should be considered as well as age.

We adjusted for age (birth year) to capture phenotypic variation in both height and the disease/biomarker outcomes (disease risk increasing with age). Age is a potential confounder of the relationship between measured height and an outcome. However, age/birth year is very unlikely to confound the relationship between height genetic variants and an outcome because these factors cannot plausibly influence genotype. Selection on height could influence allele frequencies over time leading to variation by birth year but any changes in the context of this study would be minimal.

Similarly, cohort (HUNT 2 or HUNT 3) could be a potential confounder of the phenotypic analyses but is very unlikely to confound the MR analyses. The primary focus of our manuscript is on the within-sibship MR analyses and the triangulation of evidence across different study designs. We generally found consistent effect estimates across both phenotypic and MR analyses suggesting that non-linear age and cohort effects are unlikely to have influenced our results.

At the request of the reviewers, we have added Table 1 which contains information on time periods of recruitment and other study-level information.

Is a p-value cutoff given for SNP selection into the height PGS?

We have modified the text to clarify that the height PGS was constructed using genome-wide significant variants. “The height PGS was constructed in PLINK ^52^ using 372 independent (LD clumping: 250 kb, r2 < 0.01, P < 5x10^-8^)”

Please clarify whether prevalent cases were included as well as incident cases. Given, genetics are a lifelong exposure, it might be better to include prevalent cases, so as to avoid time related biases, as explained here[1].

We have clarified in the methods that both prevalent and incident cases were included in analyses in UK Biobank (cancer/CHD) and HUNT (CHD). Cancer in HUNT is an exception as this data was extracted from questionnaires at study baseline and so only prevalent cases were included, and there was no longitudinal follow-up.

Height is largely non-modifiable from early adulthood and height genetic variants are fixed at conception, while coronary heart disease and cancer cases (both prevalent and incident) are likely to arise decades later. Therefore, we believe that the inclusion of both prevalent and incident cases to maximise power is appropriate.

ResultsPlease give a table describing the baseline characteristics of the two studies, particularly in terms of age and sex, and possibly recruitment period.

We have added a table detailing the baseline characteristics of the UK Biobank and HUNT study populations to the manuscript (Table 1).

DiscussionPlease consider the limitations of this study in more detail, specifically.1. Using a PGS for the exposure increases power but means it is difficult to check for any pleotropic effects. How might this affect interpretation of the study?

Horizontal pleiotropy is a potential bias in MR studies. Here, we focused on tackling other potential biases in MR studies (population stratification, assortative mating, and indirect genetic effects) using a within-sibship design. In theory, pleiotropy can also be evaluated in within-sibship MR. However, within family analyses generally have far lower statistical power than MR using data from unrelated individuals. Other studies have investigated whether horizontal pleiotropy can explain this relationship using unrelated individuals, these studies have generally found little evidence that horizontal pleiotropy explains these relationships. For example, see: Marouli E, Del Greco MF, Astley CM, et al. Mendelian randomisation analyses find pulmonary factors mediate the effect of height on coronary artery disease. *Communications biology.* We have added to the limitations, that with limited power, we did not perform sensitivity analyses for horizontal pleiotropy.

“A limitation of our analyses is that because of limited sibling data and the statistical inefficiency of within-family models, we have limited statistical power to investigate effects of height on disease subtypes, to further explore mechanisms using multivariable Mendelian randomization ^60^ and to perform sensitivity analyses to evaluate horizontal pleiotropy.”

2. This study is carefully designed to avoid use of overlapping samples for exposure and outcome. Is the study used for height unbiased?

We used summary statistics from the GIANT-consortium height GWAS of unrelated individuals to extract information on height SNPs. These summary data are likely to be affected by population stratification, assortative and indirect genetic effects. We included only genome-wide significant variants which, although potentially affected by these sources of associations, are still likely to be valid genetic instruments for height. Indeed, we found that the height PGS constructed from these SNPs was strongly associated with height in the within-sibship model in both studies.

3. The possibility of selection bias from survivorship. This study only includes people who have survived to recruitment, so it will be missing people who died before recruitment because of their height or height-related genetic endowment, the disease of interest or a competing risk of the disease of interest. Missing these people can attenuate or reverse estimates for exposures that affect survival. If cancer causes death at earlier ages than coronary heart disease then the results for coronary heart disease could be an artefact of the competing risk of previous death from cancer.

We have added discussion of competing risks between the two diseases to the discussion.

“An additional limitation is that our study may have been susceptible to selection and survival biases relating to non-random participation in UK Biobank and/or HUNT and the requirement of at least two siblings to survive to be recruited. Indeed, cancer and coronary heart disease are both leading worldwide causes of mortality and so cases for one disease may have reduced likelihood of developing the other disease due to increased mortality. Therefore, our study may have been susceptible to survival bias relating to competing risks. We mitigated this by defining cases for both diseases using both non-fatal and fatal events. Our study analysed families with two or more siblings jointly participating in a cohort; nevertheless, further research is required to investigate the impact of selection bias on family studies.”

4. Please clarify whether the results presented are specific to the UK Biobank and HUNT studies or explain their generalizability and transportability.

We have added to the discussion how our conclusions are based on individuals of recent European descent and so may not generalise to non-European populations if the mechanisms are context-specific.

“Further work is required to investigate if our findings generalise to non-European populations; biological mechanisms could be expected to be largely consistent across populations but context-specific (e.g., social) mechanisms could lead to geographical heterogeneity.”

5. Could differences between UK Biobank and HUNT be due to differences to design?

We have added this as a possible explanation for the heterogeneity between the studies for the biomarker analyses in the discussion.

“Additional explanations could relate to differences in biomarker measurement between studies (e.g. measuring LDL-C directly or using the Friedewald formula, differences in fasting level before samples were taken), selection bias ^64^ or differences between the cohorts in terms of recruitment and participation.”

6. Are both the comparisons between the sibling and population estimates for cancer and coronary heart disease in the expected directions?

We believe the reviewer is referring to the estimates in Figure 2. If so, the sibling and population estimates were highly consistent in terms of directionality and magnitude for both cancer and coronary heart disease.

[1] Hernán MA, Sauer BC, Hernández-Díaz S, Platt R, Shrier I: Specifying a target trial prevents immortal time bias and other self-inflicted injuries in observational analyses. J Clin Epidemiol 2016, 79:70-75.Reviewer #2 (Recommendations for the authors):How likely is it that the earlier MR results are misleading? Biased by demographics etc? It relates to the novelty of the paper because the MR association has been shown before multiple times.

As the reviewer notes, several previous MR studies have provided evidence of the height cancer and height CHD relationships using data from unrelated individuals. Similarly previous MR studies of unrelated individuals also provided consistent evidence that taller height and lower BMI increase educational attainment. However, a recent within-sibship MR study (Brumpton et al. 2020 *Nature Communications*) found that the height/BMI on education effects completely attenuate in within-sibship models suggesting that the previous MR estimates from unrelated individuals were biased. Therefore, it was certainly possible that the previous height cancer/CHD MR estimates from unrelated individuals could also have been biased.

Furthermore, previous studies in *eLife* (Berg et al., Sohail et al. 2019) also reported evidence that population stratification inflated estimates of polygenic adaptation on taller height, suggesting that height may be a phenotype particularly susceptible to bias from demographic factors. This further highlights the importance of within-sibship studies for height analyses.

What is the rationale for doing the biomarker analysis? Would it not be more informative to investigate if they mediate the effect from height on CHD?

We included analyses with biomarkers relevant to cancer and coronary heart disease to determine if height-biomarker estimates were consistent with the height-disease estimates. Biomarkers are continuous so analyses would likely have higher statistical power than for the binary disease outcomes. Statistical power was limited to perform multivariable MR analyses to investigate if these biomarkers mediate the effects of height.

Table 1 should be sample characteristics in HUNT and UKB.

We have added a table of sample characteristics for UKB and HUNT as suggested in Table 1.

What about sex stratifications? Is it possible to do for same-sex sib pairs? Is it of importance in the population at large regardless of sib ship pairs? For the traits used here it should matter.

Sex stratification would be possible in theory but would substantially reduce the sample size and statistical power because it would remove all families without same-sex siblings (e.g. a sibling pair with 1 male and 1 female) and then split the remaining dataset into two. Therefore, in practice with available data, we believe that a sex-stratified analysis would be far too underpowered to detect heterogeneity.

All figures and tables should contain information so that they are stand alone items.

We have been through the manuscript and ensured that table and figure descriptions provide enough detail. In particular, we have added descriptions for Tables 1 (new) and 2 and all of the supplemental tables.